# Longitudinal analysis of blood markers reveals progressive loss of resilience and predicts human lifespan limit

Timothy V. Pyrkov [1✉], Konstantin Avchaciov[1], Andrei E. Tarkhov[1,2,3], Leonid I. Menshikov[1,4], Andrei V. Gudkov [5,6] & Peter O. Fedichev[1,3✉]

We investigated the dynamic properties of the organism state fluctuations along individual aging trajectories in a large longitudinal database of CBC measurements from a consumer diagnostics laboratory. To simplify the analysis, we used a log-linear mortality estimate from the CBC variables as a single quantitative measure of the aging process, henceforth referred to as dynamic organism state indicator (DOSI). We observed, that the age-dependent population DOSI distribution broadening could be explained by a progressive loss of physiological resilience measured by the DOSI auto-correlation time. Extrapolation of this trend suggested that DOSI recovery time and variance would simultaneously diverge at a critical point of $120 - 150$ years of age corresponding to a complete loss of resilience. The observation was immediately confirmed by the independent analysis of correlation properties of intraday physical activity levels fluctuations collected by wearable devices. We conclude that the criticality resulting in the end of life is an intrinsic biological property of an organism that is independent of stress factors and signifies a fundamental or absolute limit of human lifespan.

[1] Gero PTE, Singapore, Singapore. [2] Skolkovo Institute of Science and Technology, Moscow, Russia. [3] Moscow Institute of Physics and Technology, Dolgoprudny, Moscow Region, Russia. [4] National Research Center 'Kurchatov Institute', Moscow, Russia. [5] Roswell Park Comprehensive Cancer Center, Elm and Carlton streets, Buffalo, NY, USA. [6] Genome Protection, Inc., Buffalo, NY, USA. ✉email: tim.pyrkov@gero.ai; peter.fedichev@gero.ai

Aging is manifested as a progressive functional decline leading to exponentially increasing prevalence[1,2] and incidence of chronic age-related diseases (e.g., cancers, diabetes, cardiovascular diseases, etc.)[3–5] and disease-specific mortality[6]. Much of our current understanding of the relationship between aging and changes in physiological variables over an organism's lifespan originates from large cross-sectional studies[7–9] and led to development of increasingly reliable "biological clocks" or "biological age" estimations reflecting age-related variations in blood markers[10], DNA methylation (DNAm) states[11,12] or patterns of locomotor activity[13–15] (see[16] for a review of biological age predictors). All-cause mortality in humans[17,18] and the incidence of chronic age-related diseases increase exponentially and double every 8 years[3]. Typically, however, the physiological indices and the derived quantities such as biological age predictions change from the levels observed in the young organism at a much lower pace than it could be expected from the Gompertzian mortality acceleration.

Most important factors that are strongly associated with age are also known as the hallmarks of aging[19] and may be, at least in principle, modified pharmacologically. In addition to that, by analogy to resilience in ecological systems, the dynamic properties such as physiological resilience measured as the recovery rate from the organism state perturbations[20,21] were also associated with mortality[22] and thus may serve as an early warning sign of impending health outcomes[23,24]. Hence, a better quantitative understanding of the intricate relationship between the slow physiological state dynamics, resilience, and the exponential morbidity and mortality acceleration is required to allow the rational design, development, and clinical validation of effective antiaging interventions.

We addressed these theoretical and practical issues by a systematic investigation of aging, organism state fluctuations, and gradual loss of resilience in a dataset involving multiple Complete Blood Counts (CBC) measured over short periods of time (a few months) from the same person along the individual aging trajectory. Neutrophil to Lymphocyte Ratio (NLR) and Red cell distribution width have been already suggested and characterized as biomarkers of aging[25–28]. Instead of focusing on individual factors, to simplify the matters, we followed[29,30] and described the organism state by means of a single variable, henceforth referred to as the dynamic organism state indicator (DOSI) in the form of the log-transformed proportional all-cause mortality model predictor. First, we observed that early in life the DOSI dynamics quantitatively follows the universal ontogenetic growth trajectory from[31]. Once the growth phase is completed, the indicator demonstrated all the expected biological age properties, such as association with age, multiple morbidity, unhealthy lifestyles, mortality and future incidence of chronic diseases.

Late in life, the dynamics of the organism state captured by DOSI along the individual aging trajectories is consistent with that of a stochastic process (random walk) on top of the slow aging drift. The increase in the DOSI variability is approximately linear with age and can be explained by the rise of the organism state recovery time. The latter is thus an independent biomarker of aging and a characteristic of resilience. Our analysis shows that the auto-correlation time of DOSI fluctuations grows (and hence the recovery rate decreases) with age from about 2 weeks to over 8 weeks for cohorts aging from 40 to 90 years. The divergence of the recovery time at advanced ages appeared to be an organism-level phenomenon. This was independently confirmed by the investigation of the variance and the autocorrelation properties of physical activity levels from another longitudinal dataset of intraday step-counts measured by wearable devices. We put forward arguments suggesting that such behavior is typical for complex systems near a bifurcation (disintegration) point and thus the progressive loss of resilience with age may be a dynamic origin of the Gompertz law. Finally, we noted, by extrapolation, that the recovery time would diverge and hence the resilience would be ultimately lost at the critical point at the age in the range of 120–150 years, thus indicating the absolute limit of human lifespan.

## Results

**Quantification of aging and development.** Complete blood counts (CBC) measurements are most frequently included in standard blood tests and thus comprise a large common subset of physiological indices reported across UKB (471473 subjects, age range 39–73 y.o.) and NHANES datasets (72,925 subjects, age range 1–85 y.o., see Supplementary Table 1 for the description of the data fields). To understand the character of age-related evolution of the organism state we employed a convenient dimensionality-reduction technique, the Principal Component Analysis (PCA). The coordinates of each point in Fig. 1A is obtained by averaging the first three Principal Component scores

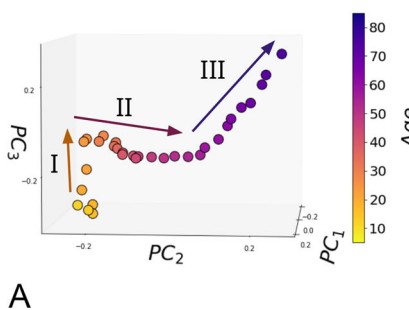
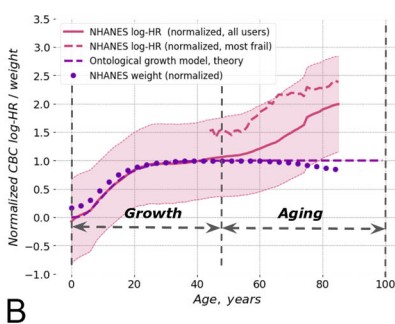
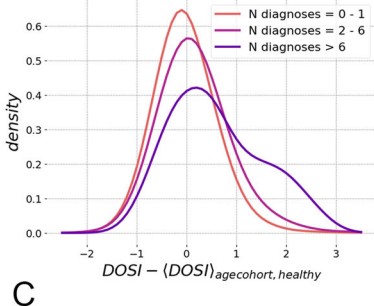

**Fig. 1 Quantification of aging and development. A** The graphical representation of the PCA for 5–85 year old NHANES participants follows an age-cohort averaged aging trajectory. Centers of each sequential age cohort are plotted in first three PCs. Three approximately linear segments are clearly seen in aging trajectory, corresponding to (I) age < 35; (II) age 35–65; (III) age > 65. **B** Dynamic organism state indicator (DOSI) mean values (solid line) and variance (shaded area) are plotted relative to age for all participants of NHANES study. The average line demonstrates nearly linear growth after age of 40. In younger ages the dependence of age is different and consistent with the universal curve suggested by the general model for ontogenetic growth[31]. To illustrate the general character of this early-life dependence we superimposed it with the curve of mean weight in age cohorts of the same population (dotted line). All values are plotted in normalized from as in[31]. The average DOSI of the "most frail" ("compound morbidity index", CMI > 0.6) individuals is shown with the dashed line. **C** Distributions of sex- and age-adjusted DOSI in cohorts of NHANES participants in different morbidity categories relative to the DOSI mean in cohorts of "non-frail" (1 or no diagnoses, CMI < 0.1) individuals. Note that the distribution function in the "most frail" group (more than six diagnoses, CMI > 0.6) exhibited the largest shift and a profound deviation from the symmetric form.

of PCA-transformed CBC variables in age-matched cohorts in NHANES dataset. The average points follow a well-defined trajectory or a flow in the multivariate configuration space spanned by the physiological variables and clearly correspond to various stages of the organism development and aging.

Qualitatively, we differentiated three distinctive segments of the aging trajectory, corresponding to (I) early adulthood (16–35 y.o.); (II) middle ages (35–65 y.o.); and (III) older ages (older than 65 y.o.). The middle segment in trajectories for women has additional change of direction presumably associated with menopause, but we leave its investigation for future work. Inside each of the segments, the trajectory was approximately linear. This suggests that over long periods of time (age), CBC variations other than noise could be described by the dynamics of a single dynamic variable (degree of freedom) tracking the distance travelled along the aging trajectory and henceforth referred to as the DOSI.

Morbidity and mortality rates increase exponentially with age and a log-linear risk predictor model is a good starting point for characterization of the functional state of an organism and quantification of the aging process[15,29]. Accordingly, we employed Cox proportional hazards model[32] and trained it using the death register of the NHANES study using log-transformed CBC measurements and sex variable (but not age) as covariates. Altogether, the training subset comprised participants aged 40 y.o. and older. The mortality risk model yielded a single value of log-hazards ratio for every subject and increased in full age range of NHANES participants (Fig. 1B). As we will see below, it was a useful and dynamic measure of the organism state henceforth identified with DOSI.

Early in life the dynamics of the organism state has, of course, nothing to do with the late-life increase of mortality rate (i.e., aging), but is rather associated with ontogenetic growth. Accordingly, we checked that the organism state measured by DOSI follows closely the theoretical trajectory of the body mass adopted from[31]:

$$x(t) = X \left( 1 - \left[ 1 - \left( \frac{x_0}{X} \right)^{\frac{1}{4}} \right] e^{\frac{-t}{t_0}} \right)^4 . \tag{1}$$

Here $x$ is the body mass, or in the linear regime any quantity such as DOSI depending on the body mass, $t$ is the age, $t_0$ is the characteristic time scale associated with the development, and $x_0$ and $X$ are the asymptotic levels of same property at birth and in the fully grown state, respectively. The dots and the dashed lines in Fig. 1B represents age-cohorts averaged body mass trajectory and the best fit of the age-cohort averaged DOSI levels by Eq. (1) for the same NHANES participants. The approximation works remarkably well up until the age of about 40. The characteristic time scale from the fit, $t_0 = 6.8$ years, coincides almost exactly with the best fit value of 6.3 years obtained from the fit of body mass trajectory.

As the body size increases, the metabolic output per unit mass slows down and the organism reaches a steady state corresponding to the fully grown organism. The inspection of Fig. 1B shows, however, that the equilibrium solution of the organism growth problem appears to be unstable in the long run and the organism state dynamics measured by DOSI exhibits deviations from the stationary solution beyond the age of ~40 years old.

To separate the effects of chronic diseases from disease-free aging, we followed[33] and characterized the health status of each study participant based the number of health conditions diagnosed for an individual normalized to the total number of conditions included in the analysis to yield the "compound morbidity index" (CMI) with values ranging from 0 to 1. The list of health conditions common to the NHANES and UKB studies

that were used for CMI determination is given in Supplementary Table 2.

CMI can be viewed as a convenient proxy to the Frailty Index introduced in[34], that is a composite marker, depending on the prevalence of 46 health deficits. Unlike the Frailty Index, CMI requires only the variables that are available simultaneously in NHANES and UKB. In NHANES, among individuals aged 40 and older, the correlation between the Frailty Index and CMI was pretty high (Pearson $r = 0.64$). Therefore we accept semi-quantitative correspondence between CMI and Frailty Index and categorize UKB and NHANES participants in cohorts of individuals of increasing level of frailty according to CMI.

Multiple morbidity manifests itself as elevated DOSI levels. This can be readily seen from the difference between the solid and dashed lines in Fig. 1B, which represent the DOSI means in the cohorts of healthy ("non-frail", CMI < 0.1) and "most frail" (CMI > 0.6) NHANES participants, respectively. In groups stratified by increasing number of health condition diagnoses, the normalized distribution of DOSI values (after adjustment by the respective mean levels in age- and sex-matched cohorts of healthy subjects) exhibited a progressive shift and increased variability (see Fig. 1C and Supplementary Fig. 1b for NHANES and UKB, respectively).

For both NHANES and UKB, the largest shift was observed in the "most frail" (CMI > 0.6) population. The increasingly heavy tail at the high end of the DOSI distribution in this group is characteristic of an admixture of a distinct group of individuals occupying the adjacent region in the configuration space corresponding to the largest possible DOSI levels. Therefore, DOSI displacement from zero-mean (after proper adjustments for age and sex) was expected to reflect the fraction of "most frail" individuals in a cohort of any given age. This was confirmed to be true using the NHANES dataset (Fig. 2A; $r = 0.83$).

The fraction of "most frail" subjects still alive increased exponentially at every given age until the age corresponding to the end of healthspan was reached. The characteristic doubling rate constants for the "most frail" population fractions were 0.087 and 0.094 per year in the NHANES and the UKB cohorts, respectively, in comfortable agreement with the accepted Gompertz mortality doubling rate of 0.085 per year[35], see Fig. 2B.

We note that the prevalence of diseases in the NHANES cohort is consistently higher than that in the UKB population, although the average lifespan is comparable in the two countries. This may be a consequence of the enrollment bias in the UKB: life tables analysis in[36] suggests the UKB subjects appear to outlive typical UK residents.

## Dynamic organism state indicator (DOSI) and health risks.
In the most healthy subjects, i.e., those with no diagnosed diseases at the time of assessment, the DOSI predicted the future incidence of chronic age-related diseases observed during 10-year follow-up in the UB study (Supplementary Table 2). There was no relevant information available in NHANES. We tested this association using a series of Cox proportional hazard models trained to predict the age at the onset/diagnosis of specific diseases. We observed that the morbidity hazard ratios associated with the DOSI relative to its mean in age- and sex-matched cohorts were statistically significant predictors for at least the most prevalent health conditions (those with more than 3000 occurrences in the UKB population). The effect size (HR ≈ 1.03–1.07) was the same regardless of whether a disease was diagnosed first in a given individual or followed any number of other diseases. Only emphysema and heart failure which are known to be strongly associated with increased neutrophil counts[37,38] demonstrated particularly high associations. Therefore, we conclude that the DOSI is a characteristic of overall health status that is universally

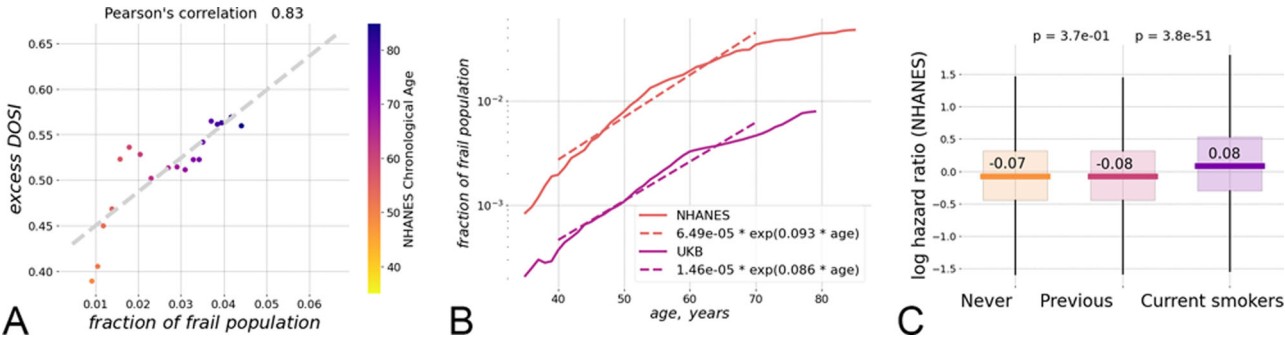

**Fig. 2 The relation between the dynamic organism state indicator (DOSI) and lifestyles, frailty, and health risks. A** Fraction of frail persons is strongly correlated with the excess DOSI levels, that is the difference between the DOSI of an individual and its average and the sex- and age-matched cohort in the "non-frail" population in NHANES. **B** Exponential fit showed that until the age of 70 y.o. the fraction of the "most frail" individuals in the population grows approximately exponentially with age with the doubling rate constants of 0.08 and 0.10 per year in the UKB and the NHANES cohorts, respectively. **C** Distribution of log-hazards ratio in age- and sex-matched cohorts of NHANES participants who never smoked, smoked previously but quit prior to the time of study participation, or were current smokers at the time of the study. The DOSI level is elevated for current smokers, while it is almost indistinguishable between never-smokers and those who quit smoking (two-sided Mann–Whitney test $p > 0.05$). Each boxplot shows the center (median) of the distribution, boxplot bounds show the 25 and 75% percentiles and boxplot whiskers show the 5 and 95% percentiles.

associated with the risks of developing the most prevalent diseases and, therefore, with the end of healthspan as indicated by the onset of the first morbidity (HR ≈ 1.05 for the "First morbidity" entry in Supplementary Table 2).

In the most healthy "non-frail" individuals with life-shortening lifestyles/behaviors, such as smoking, the DOSI was also elevated, indicating a higher level of risks of future diseases and death (Fig. 2C). Notably and in agreement with the dynamic nature of DOSI, the effect of smoking appeared to be reversible: while the age- and sex- adjusted DOSI means were higher in current smokers compared to non-smokers, they were indistinguishable between groups of individuals who never smoked and who quit smoking (c.f.[15,39]).

**Physiological state fluctuations and loss of resilience**. To understand the dynamic properties of the organism state fluctuations in relation to aging and diseases, we used two large longitudinal datasets, jointly referred to and available as GERO-LONG, including anonymized information on: (a) CBC measurements from InVitro, the major Russian clinical diagnostics laboratory and (b) physical activity records measured by step counts collected by means of a freely available iPhone application.

The CBC slice of the combined dataset included blood test results from 388 male and 694 female subjects aged 30–90 with complete CBC analyses that were sampled 10–20 times within a period of more than 3 years (up to 42 months).

There was no medical condition information available for the GEROLONG subjects. Hence, for the CBC measurements we used the mean DOSI level corresponding to the "most frail" NHANES and UKB participants as the cutoff value to select "non-frail" GEROLONG individuals (141 male and 266 female subjects aged 40–90) for subsequent analysis.

The difference between the mean DOSI levels in groups of the middle-aged and the eldest available individuals was of the same order as the variation of DOSI across the population at any given age (see Fig. 1B). Accordingly, serial CBC measurements along the individual aging trajectories revealed large stochastic fluctuations of the physiological variables around its mean values, which were considerably different among individual study participants. Naturally, physiological variables at any given moment of time reflect a large number of stochastic factors, such as manifestation of the organism responses to endogenous and external factors (as in Fig. 2C). We therefore focused on the statistical properties of the organism state fluctuations.

Auto-correlation function is the single most important statistical property of a stationary stochastic process represented by a time series $x(t)$:

$$C(\Delta t) = \langle \delta x(t + \Delta t) \delta x(t) \rangle_t, \qquad (2)$$

where $\Delta t$ is the time lag between the subsequent measurements of $x$, $\delta x(t) = x(t) - \langle x \rangle_t$ is the deviation of $x$ from its mean value produced by the averaging $\langle x(t) \rangle_t$ along the individual trajectory (see e.g.,[40]).

The autocorrelation function of $x = $ DOSI averaged over individual trajectories in subsequent age cohorts of GEROLONG dataset was plotted vs. the delay time in Fig. 3A and exhibited exponential decay over a time scale of ~2–8 weeks depending on age.

The exponential character of the autocorrelation function, $C(\Delta t) \sim \exp(-\varepsilon \Delta t)$ is a signature of stochastic processes following a simple Langevin equation:

$$\delta \dot{x} = -\varepsilon \delta x + f(t), \qquad (3)$$

where $\delta \dot{x}$ stands for the rate of change in fluctuations $\delta x$, $\varepsilon$ is the relaxation or recovery rate, and $f$ is the "force" responsible for deviation of the organism state from its equilibrium.

The auto-correlation function decay time (or simply the auto-correlation time) is inversely proportional to the relaxation (recovery) rate $\varepsilon$ and characterizes the time scale involved in the equilibration of a system's state in response to external perturbations. We therefore propose using this quantity as a measure of an organism's "resilience", the capacity of an individual organism to resist and recover from the effects of physiological or pathological stresses[41,42].

We fitted the DOSI auto-correlation functions averaged over individuals representing subsequent age-matched cohorts to an exponential function of the time delay. We observed that recovery rates obtained from fitting to data in the subsequent age-cohorts decreased approximately linearly with age (Fig. 3C). Extrapolation to older ages suggested that the equilibration rate and hence the resilience is gradually lost over time and is expected to vanish (and hence the recovery time to diverge), at some age of ~120–150 y.o.).

The exponential decay of auto-correlation function is not merely a peculiarity of an organism state indicator computed from CBC. We were able to use another set of high resolution longitudinal measurements of daily step counts collected by wearable devices. Step counts measurements were obtained from

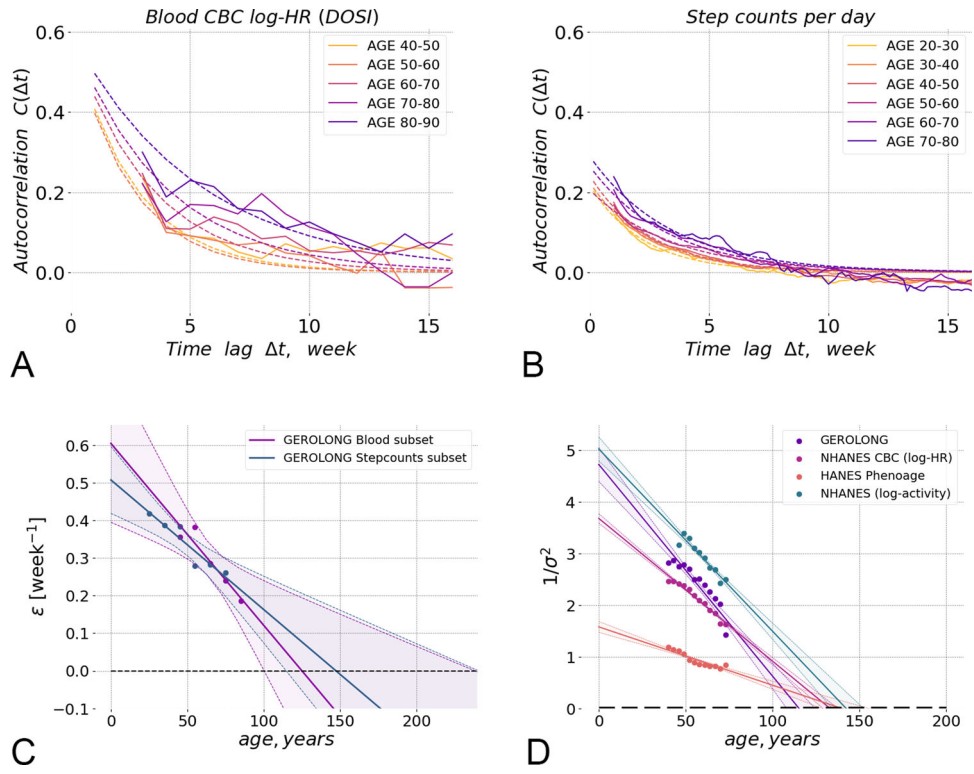

**Fig. 3 Physiological state fluctuations and loss of resilience. A** The auto-correlation function $C(\Delta t)$ of the Dynamic organism state indicator (DOSI) fluctuations during several weeks averaged in sequential 10-year age-cohorts of GEROLONG subjects showed gradual age-related remodelling. Experimental data and fit to autocorrelation function are shown with solid and dashed lines, respectively. The DOSI correlations are lost over time $\Delta t$ between the measurements and, hence, the DOSI deviations from its age norm reach the equilibrium distribution faster in younger individuals. **B** The auto-correlation function $C(\Delta t)$ of fluctuations of the negative logaritm of steps-per-day during several weeks averaged in sequential 10-year age-cohorts of GEROLONG Stepcounts subset subjects showed similar gradual age-related remodelling. **C** The DOSI relaxation rate (or the inverse characteristic recovery time) computed for sequential age-matched cohorts from the GEROLONG dataset decreased approximately linearly with age and could be extrapolated to zero at an age in the range of ~110–170 y.o. (at this point, there is complete loss of resilience and, hence, loss of stability of the organism state). The solid lines and shaded areas show the line of linear regression fit and its 95% confidence interval. **D** The inverse variance of DOSI decreased linearly in all investigated datasets and its extrapolated value vanished (hence, the variance diverged) at an age in the range of 120–150 y.o. We performed the linear fit for subjects 40 y.o. and older, excluding the "most frail" ("compound morbidity index", CMI > 0.6) individuals. The solid lines and shaded areas show the line of linear regression fit and its 95% confidence interval. The blue dots and lines show the inverse variance of log-scaled measure of total physical activity (the number of steps per day recorded by a wearable accelerometer) for NHANES participants. Phenoage[29], calculated using explicit age and additional blood biochemistry parameters also demonstrated age-related decrease of the inverse variance in NHANES population.

users of fitness wristband (3032 females, 1783 males of age 20–85 y.o.). The number of measurements for each user was at least 30 days and up to 5 years.

In[15] we observed, that the variability of physical activity (namely, the logarithm of the average physical activity), that is another hallmark of aging and is associated with age and risks of death or major deceases, also increases with age and hence may be used as an organism state indicator. The autocorrelation function of the physical activity levels shows already familiar exponential profile and signs of the loss of resilience in subsequent age-matched cohorts as shown in Fig. 3B.

The recovery rate inferred from as the inverse autocorrelation time from physical activity levels trajectories is plotted alongside the recovery rates from CBC-derived DOSI in Fig. 3C. We observed that the recovery rates revealed by the organism state fluctuations measured in apparently unrelated subsystems of the organism (the blood cell counts and physical activity levels) are highly concordant, both decrease as the function of the chronological age at the same pace, and, if the extrapolation holds, vanish at the same limiting age.

Eq. (3) predicts, that the variance of DOSI should also increase with age. Indeed, according to the solution of the Langevin equation with a purely random and uncorrelated force,

$\langle f(t + \Delta t)f(t) \rangle_t = B\delta(\Delta t)$ (with $\delta(x)$ being the Dirac's delta-fucntion and $B$ being the power of the stochastic noise), the fluctuations of $x =$ DOSI should increase with age thus reflecting the dynamics of the recovery rate: $\sigma^2 \equiv \langle \delta x^2 \rangle \sim B/\varepsilon$.

Remarkably, the variability in a DOSI did increase with age in every dataset evaluated in this study. Following our theoretical expectations of the inverse relation between the resilience and the fluctuations, we plotted the inverse variance of the DOSI computed in sex- and age-matched cohorts representing the most healthy subjects (see Fig. 3D). Again, extrapolation suggested that, if the tendency holds at older ages, the population variability would increase indefinitely at an age of ~120–150 y.o.

As expected, the amplification of the fluctuations of the organism state variables with age is not limited to CBC features. In Fig. 3D we plotted the inverse variance of this physical activity feature and found that it linearly decreases with age in such a way that the extrapolated variance diverges at the same critical point at the age of ~120–150 y.o.

To demonstrate the universality of of the organism state dynamics, we followed the fluctuation properties of the Phenoage, another log-linear mortality predictor trained using the explicit age, sex and a number of biochemical blood markers[29]. By its nature, PhenoAge is another DOSI produced from a different set

of features. Unfortunately, we could not not obtain a sufficient number of individuals with all the relevant markers measurements from the longitudinal dataset from InVitro. Accordingly, we could not compute the corresponding autocorrelation function. We were, however, able to compute PhenoAge for NHANES subjects and observed an increase in variability of the PhenoAge estimate as a function of chronological age and a possible divergence of PhenoAge fluctuations at around the age of 150 y.o.

## Discussion

The simultaneous divergence of the organism state recovery times (critical slowing down in Fig. 3C) and the increasing dynamic range of the the organism state fluctuations (critical fluctuations in Fig. 3D) observed independently in two biological signals is characteristic of proximity of a critical point[23,40] at some advanced age over 100 y.o. Under these circumstances, the organism state dynamics are stochastic and dominated by the variation of the single dynamic variable (also known as the order parameter) associated with criticality[23,43]. A proper identification of such a feature requires massive high-quality longitudinal measurements and sophisticated approaches auto-regressive models. In a similar study involving CBC variables of aging mice, we were able to obtain an accurate predictor associated with the age, risks of death (and the remaining lifespan), and frailty[44]. In this work we turned the reasoning around and choose to quantify the organism state by the log-linear proportional hazards estimate of the mortality rate followed[15,29,45], using CBC and physical activity variables. This inherently dynamic quantitative organism state indicator (DOSI) increased with age, predicted the prospective incidence of age-related diseases and death, and was elevated in cohorts representing typical life-shortening lifestyles, such as smoking, or exhibiting multiple morbidity.

The log-linear risks model predictor demonstrated a non-trivial dependence on age also early in life, that is in the age range with almost no recorded mortality events in the training dataset. The age-cohort averaged DOSI increased and then reached a plateau (Fig. 1B) in good quantitatively consistent with the predictions of the universal theory of ontogenetic growth[31]. The agreement between the theory and the DOSI dependence on age is very good, and hence we are led to believe that the features of the "aging trajectory" in Fig. 1A are not coincidental artifacts of data analysis.

According to the theory, the development of any organism is the result of a competition between the production of new tissue and life-sustaining activities. The total amount of the energy available scales as the fractional power of the body mass $m^{3/4}$ according to the universal allometric Kleiber–West law[46,47]. On the one hand, the energy requirements for the organism maintenance increase linearly as the body mass grows and hence the initial excess metabolic power drives the growth of the organism until it reaches the dynamic equilibrium corresponding to the mature animal state.

As we can see in Fig. 1B, the mature human organism is dynamically unstable in the long run and deviations from the ontogenetic growth theory predictions pick up slowly well after the organism is fully formed. The organism state dynamics measured by DOSI over lifetime qualitatively reveals at least three regimes reflecting growth, maturation, and aging, respectively. The apparent life-stages correspond well to the results of multi-variate PCA of CBC variance (Fig. 1A) in this work and also that of physical activity acceleration/deceleration patterns from[15]. Every arm of the aging trajectory is characterized by a specific set of features strongly associated with age in the signal.

Schematically, the reported features of the longitudinal organism state dynamics can be summarized with the help of the

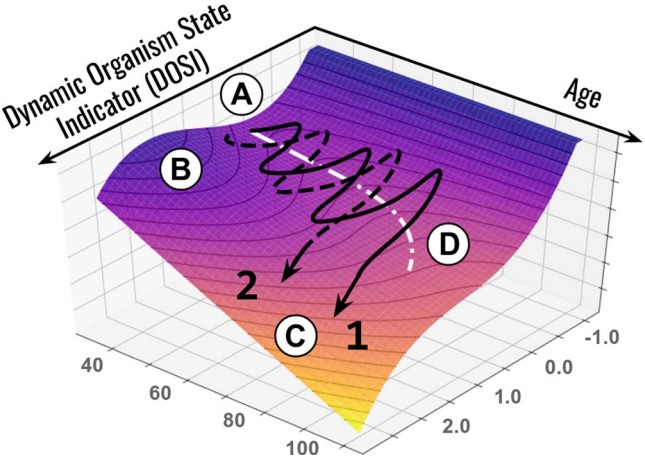

**Fig. 4 Schematic representation of loss of resilience along aging trajectories.** Representative aging trajectories are superimposed over the potential energy landscape (vertical axis) representing regulatory constraints. The stability basin (**A**) is separated from the unstable region (**C**) by the potential energy barrier (**B**). Aging leads to a gradual decrease in the activation energy and barrier curvature and an exponential increase in the probability of barrier crossing. The stochastic activation into a dynamically unstable (frail) state is associated with acquisition of multiple morbidities and certain death of an organism. The white dotted line (**D**) represents the trajectory of the attraction basin minimum. Examples 1 (black solid line) and 2 (black dashed line) represent individual life-long stochastic DOSI trajectories that differ with respect to the age of first chronic disease diagnosis.

following qualitative picture (Fig. 4). Far from the critical point (at younger ages), the organism state perturbations can be thought of as confined to the vicinity of a possible stable equilibrium state in a potential energy basin (A). Initially, the dynamic stability is provided by a sufficiently high potential energy barrier (B) separating this stability basin from the inevitably present dynamically unstable regions (C) in the space of physiological parameters. While instability basin, an organism state experiences stochastic deviation from the metastable equilibrium state, which is gradually displaced (see the dashed line D) in the course of aging even for the successfully aging individuals.

The characteristic organism state auto-correlation time demonstrated here (3–6 weeks, see Fig. 3A) is much shorter than lifespan. The dramatic separation of time scales makes it very unlikely that the linear decline of the recovery force measured by the recovery rate in Fig. 3C can be explained by the dynamics of the organism state captured by the DOSI variation alone. Therefore, we conclude that the progressive remodeling of the attraction basin geometry reflects adjustment of the DOSI fluctuations to the slow independent process that is aging itself. In this view, the aging drift of the DOSI mean in cohorts of healthy individuals (as in Fig. 1B) is the adaptive organism-level response reflecting, on average, the increasing stress produced by the aging process.

The longitudinal analysis in this work demonstrated that the organism state measured by DOSI follows a stochastic trajectory driven mainly by the organism responses to unpredictable stress factors. Over lifetime, DOSI increases slowly, on average. The dynamic range of the organism state fluctuations is proportional to the power of noise and is inversely proportional to the recovery rate of the DOSI fluctuations. Therefore, the organism state of healthy individuals at any given age is described by the mean DOSI level, the DOSI variability and its auto-correlation time. Together, the three quantities comprise the minimum set of bio-markers of stress and aging in humans and could be determined

and altered, in principle, by different biological mechanisms and therapeutic modalities.

The DOSI recovery rate characterizes fluctuations of DOSI on time scales from few weeks to few months, decreases with age and thus indicates the progressive loss of physiological resilience. Such age-related remodeling of recovery rates has been previously observed in studies of various physiological and functional parameters in humans and other mammals. For example, in humans, a gradual increase in recovery time required after macular surgery was reported in sequential 10-year age cohorts[48] and age was shown to be a significant factor for twelve months recovery and the duration of hospitalization after hip fracture surgery[49,50], coronary artery bypass[51], acute lateral ankle ligament sprain[52]. A mouse model suggested that the rate of healing of skin wounds also can be a predictor of longevity[53].

The resilience can only be measured directly from high-quality longitudinal physiological data. The Framingham Heart Study[7], Dunedin Multidisciplinary Health and Development Study[54] and other efforts produced a growing number of reports involving statistical analysis of repeated measurements from the same persons, see, e.g.,[55,56]. Most of the time, however, the subsequent samples are years apart and hence time between the measurements greatly exceeds the organism state autocorrelation time reported here. This is why, to the best of our understanding, the relation of the organism state recovery rate and mortality has remained largely elusive.

In the presence of stresses, the loss of resilience should lead to destabilization of the organism state. Indeed, in a reasonably smooth potential energy landscape forming the basin of attraction, the activation energy required for crossing the protective barrier ($B$) decreases along with the curvature at the same pace, that is, linearly with age. Whenever the protective barrier is crossed, dynamic stability is lost (see example trajectories 1 and 2 in Fig. 4, which differ by the age of crossing) and deviations in the physiological parameters develop beyond control, leading to multiple morbidities, and, eventually, death.

On a population level, activation into such a frail state is driven by stochastic forces and occurs approximately at the age corresponding to the end of healthspan, understood as "disease-free survival". Since the probability of barrier crossing is an exponential function of the required activation energy (i.e., the barrier height)[40], the weak coupling between DOSI fluctuations and aging is then the dynamic origin of exponential mortality acceleration known as the Gompertz law. Since the remaining lifespan of an individual in the frail state is short, the proportion of frail subjects at any given age is proportional to the barrier crossing rate, which is an exponential function of age (see Fig. 2B).

The end of healthspan can therefore be viewed as a form of a nucleation transition[40], corresponding in our case to the spontaneous formation of states of chronic diseases out of the metastable phase (healthy organisms). The DOSI is then the order parameter associated with the organism-level stress responses at younger ages and plays the role of the "reaction coordinate" of the transition to the frail state later in life. All chronic diseases and death in our model originate from the dynamic instability associated with single protective barrier crossings. This is, of course, a simplification and yet the assumption could naturally explain why mortality and the incidence of major age-related diseases increase exponentially with age at approximately the same rate[3].

The reduction of slow organism state dynamics to that of a single variable is typical for the proximity of a tipping or critical point[23]. DOSI is therefore the property of the organism as a whole, rather than a characteristics of any specific functional subsystem or organism compartment. We did observe a neat concordance between the decrease in the organism state recovery rates (Fig. 3C) and DOSI variance divergence (Fig. 3D) from

seemingly unrelated sources such as blood markers and the physical activity variables. This is likely a manifestation of common dynamic origin of a substantial part of fluctuations in diverse biological signals ranging from blood markers (CBC and PhenoAge covariates) to physical activity levels. We therefore predict that similar divergence of variance and increase in auto-correlation times will be found in future studies involving other risk-associated markers, including DNAm clocks.

According to the presented model, early in life the dynamics of DOSI is described by a simple Langevin Eq. (3). External stresses (such as smoking) or diseases produce perturbations that modify the shape of the effective potential leading to the shift of the equilibrium DOSI position. For example, the mean DOSI values in cohorts of individuals who never smoked or who quit smoking are indistinguishable from each other, yet significantly different from (lower than) the mean DOSI in the cohort of smokers (Fig. 2C). Thus, the effect of the external stress factor is reflected by a change in the DOSI and is reversed as soon as the factor is removed.

These findings agree with earlier observations suggesting that the effects of smoking on remaining lifespan and on the risks of developing diseases are mostly reversible once smoking is ceased well before the onset of chronic diseases[15,39]. The decline in the lung cancer risk after smoking ablation[57] is slower than the recovery rate reported here. This may be the evidence suggesting that long-time stresses may cause hard-to repair damage to the specific tissues and thus produce lasting effects on the resilience.

In the absence of chronic diseases when the organism state is dynamically stable, the elevation of physiological variables associated with the DOSI indicates reversible activation of the most generic protective stress responses. Moderately elevated DOSI levels are therefore protective responses that can measured by molecular markers (e.g., C-reactive protein) and affects general physical and mental health status[45]. We also predict, that death is preempted by the activation into a state with excess DOSI and loss of resilience. The excessive DOSI levels observed in older individuals can be thought of as an aberrant activation of stress-responses beyond the dynamic stability range. Thus elevated levels and long auto-correlation times of DOSI fluctuations are therefore characteristics of chronic diseases and predict death.

We propose that therapies targeting frailty-associated phenotypes (e.g., inflammation) would, therefore, produce distinctly different effects in disease-free vs. frail populations. In healthy subjects, who reside in the region of the stability basin ($B$) (see Fig. 4), a treatment-induced reduction of DOSI would quickly saturate over the characteristic auto-correlation time and lead to a moderate decrease in long-term risk of morbidity and death without a change in resilience. Technically, this would translate into an increase in healthspan, although the reduction of health risks would be transient and disappear after cessation of the treatment. In frail individuals, however, the intervention could produce lasting effects and reduce frailty, thus increasing lifespan beyond healthspan. This argument may be supported by longitudinal studies in mice suggesting that the organism state is dynamically unstable, the organism state fluctuations get amplified exponentially at a rate compatible with the mortality rate doubling time, and the effects of transient treatments with life-extending drugs such as rapamycin produce a lasting attenuation of frailty index[44].

The emergence of chronic diseases out of increasingly unstable fluctuations of the organism state provides the necessary dynamic argument to support the derivation of the Gompertz mortality law in the Strehler–Mildvan theory of aging[58]. In[59,60], the authors suggested that the exponential growth of disease burden observed in the National Population Health Survey of Canadians over 20 y. o. could be explained by an age-related decrease in organism recovery in the face of a constant rate of exposure to environmental stresses.

Our study provides evidence suggesting that late in life the organism state dynamics is dominated by features that originate from the proximity of the critical point, corresponding to the vanishing resilience. The exact parameters, such as maximum lifespan, are the results of extrapolations yielding the estimate in the range of 100–150 years. The questions of whether the critical point corresponds to a specific age or even achievable along a realistic trajectory, are not too practical: due to the presence of strong stochastic forces, most individuals escape the attraction basin, lose the resilience and disintegrate into states corresponding to chronic diseases well before reaching the ultimate age. Hence the extrapolation may serve to establish the upper bound on attainable age or the limiting lifespan.

We therefore argue, that the loss of resilience cannot be avoided even in the most successfully aging individuals and, therefore, could explain the very high mortality seen in cohorts of super-centennials characterized by the so-called compression of morbidity (late onset of age-related diseases[61]). Formally, such a state of "zero-resilience" at the critical point corresponds to the absolute zero on the vitality scale in the Strehler–Mildvan theory of aging, thus representing a natural limit on human lifespan. We also note, that very late in life, as the probability of the loss of resilience increases, so should the deviations from Gompertz mortality law. A recent careful analysis of human demographic data supports this argument and yields an estimate for limiting lifespan of 138 years[62].

The semi-quantitative description of human aging and morbidity proposed here should work well long before the maximum age and belongs to a class of phenomenological models. Whereas it is possible to associate the variation of the organism state measured by DOSI with the effects of stresses or diseases, the data analysis presented here does not provide any mechanistic explanations for the progressive loss of resilience. It is worth to note that the recent study predicts the maximum human lifespan limit from telomere shortening[63] that is compatible with the estimations presented here. It would therefore be interesting to see if the resilience loss in human cohorts is associated or even caused by the loss of regenerative capacity due to Hayflick limit.

The proximity of the critical point revealed in this work indicates that the apparent human lifespan limit is not likely to be improved by therapies aimed against specific chronic diseases or frailty syndrome. Thus, no dramatic improvement of the maximum lifespan and hence strong life extension is possible by preventing or curing diseases without interception of the aging process, the root cause of the underlying loss of resilience. We do not foresee any laws of nature prohibiting such an intervention. Therefore, further development of the aging model presented in this work may be a step toward experimental demonstration of a dramatic life-extending therapy.

## Methods

**Complete blood count datasets**. NHANES CBC data were retrieved from the category "Complete Blood Count with 5-part Differential - Whole Blood" of Laboratory data for NHANES surveys 1999–2014. Corresponding UKB CBC data fields with related database codes are listed in Supplementary Table 1. The fraction of samples with missing (or filled with zero) CBC data was <0.035% in any studied dataset and those samples were discarded. Differential white blood cell percentages were converted to cell counts by multiplication by $0.01 \times$ White blood ceel count. All CBC parameters were log-transformed and normalized to zero-mean and unit-variance based on data of NHANES participants aged 40 y.o. and older to further carry out PCA and train Cox proportional hazards model.

**Step counts datasets**. NHANES step counts per minute records during 1 week were retrieved from the category "Physical Activity Monitor" of Examination data for NHANES 2005–2006 survey. Autocorrelation of log-transformed daily step counts was calculated using data from "Fitbit" devices of 4532 users aged 20–80 y.o. (1601 male and 2892 female).

**Hazards model**. The Cox proportional hazards model was trained using NHANES 2015 Public-Use Linked Mortality data. We used CBC data and mortality linked follow-up available for 40,592 NHANES participants aged 18–85 y.o.. NHANES population aged 40–85 y.o. was split randomly into training (12,851 participants) and test (12,883 participants) subsets. Cox model was trained using training subset (6259 male and 6592 female) with 2392 recorded death events during follow-up until the year 2015 (1999–2014 surveys). CBC components and the biological sex label were used as covariates.

The model predicted the all-cause mortality well and yielded a concordance index value of $CI = 0.68$ and $CI = 0.67$ in NHANES training and test subsets and $CI = 0.65$ in UKB (samples collected 2007–2011, 218,530 male and 257,965 female participants aged 39–75 y.o., 28,210 recorded death events during follow-up until the year 2020). The Cox proportional hazards model was used as implemented in lifelines package (version 0.25.1) in python. The model was then applied to calculate the hazards ratio for all samples in the GEROLONG, UKB and NHANES cohorts (including individuals younger than 40 y.o.).

The DOSI defined as log-hazard ratio of the risk model throughout the manuscript) turned out to be equally well associated with mortality in the NHANES study ($HR = 1.43$) used for training of the risk model and in the independent UKB study ($HR = 1.35$; Supplementary Table 2), which was used as a validation dataset.

All data analyses were carried out in python 3.8 scripts using libraries NumPy (version 1.18.5), SciPy (version 1.5.2) and Lifelines (version 0.25.1).

**The most prevalent chronic diseases and health status**. We quantified the health status of individuals using the sum of major age-related medical conditions (MCQ) that they were diagnosed with, which we termed the CMI. The CMI is similar in spirit to the frailty index suggested for NHANES[33]. We were not able to use the frailty index because it was based on Questionnaire and Examination data that were not consistent between all NHANES surveys. Also, we did not have enough corresponding data for the UKB dataset. For CMI determination, we followed[61] and selected the top 11 morbidities strongly associated with age after the age of 40. The list of health conditions included cancer (any kind), cardiovascular conditions (angina pectoris, coronary heart disease, heart attack, heart failure, stroke, or hypertension), diabetes, arthritis, and emphysema. Notably, we did not include dementia in the list of diseases since it occurs late in life and hence is severely underrepresented in the UKB cohort due to its limited age range. We categorized participants who had more than 6 of those conditions as the "most frail" ($CMI > 0.6$), and those with $CMI < 0.1$ as the "non-frail". NHANES data for diagnosis with a health condition and age at diagnosis is available in the questionnaire category "MCQ". Data on diabetes and hypertension was retrieved additionally from questionnaire categories "Diabetes" (DIQ) and "Blood Pressure & Cholesterol", respectively.

UK Biobank does not provide aggregated data on these MCQ. Rather, it provides self-reported questionnaire data (UKB, Category 100074) and diagnoses made during hospital in-patient stay according to ICD10 codes (UKB, Category 2002). We aggregated self-reported and ICD10 (block level) data to match that of NHANES for transferability of the results between populations and datasets. We used the following ICD10 codes to cover the health conditions in UK Biobank: hypertension (I10-I15), arthritis (M00-M25), cancer (C00-C99), diabetes (E10-E14), coronary heart disease (I20-I25), myocardial infarction (I21, I22), angina pectoris (I20), stroke (I60-I64), emphysema (J43, J44), and congestive heart failure (I50).

Consistently with our previous observations in the NHANES and UKB cohorts, DOSI also increased with age in the longitudinal GEROLONG cohort. The average DOSI level as well as its population variance at any given age were, however, considerably larger than those in the reference "non-frail" groups from the NHANES and UKB studies (see Supplementary Fig. 1a). This difference likely reflects an enrollment bias: many of the GEROLONG blood samples were obtained from patients visiting clinic centers, presumably due to health issues. This could explain why the GEROLONG population appeared generally more frail in terms of DOSI than the reference cohorts of the same age from other studies (Supplementary Fig. 1a, compare the relative positions of the solid blue line and the two dashed lines representing the GEROLONG cohort and the frail cohorts of the NHANES and UKB studies, respectively).

**Reporting summary**. Further information on research design is available in the Nature Research Reporting Summary linked to this article.

## Data availability

The data that support the findings of this study are available at the NHANES web-site https://www.cdc.gov/nchs/nhanes, at UK Biobank data access procedure described at https://www.ukbiobank.ac.uk/enable-your-research. Additional data are available from the corresponding authors on reasonable request.

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

## Acknowledgements

This research has been conducted using data from UK Biobank, a major biomedical database (UK Biobank website: www.ukbiobank.ac.uk; UK Biobank project ID 21988).

## Author contributions

T.V.P., L.I.M., A.V.G., and P.O.F. designed the study and analyzed the results. T.V.P., K.A., A.E.T. and P.O.F. performed calculations and data analysis. All authors discussed the results, wrote and reviewed the paper.

## Competing interests

P.O.F. is a shareholder of Gero PTE. A.G. is a member of Gero PTE Advisory Board. T.V.P., A.E.T., K.A., L.I.M., and P.O.F. are employees of Gero PTE. The study was funded by Gero PTE.
