## [Peer Review File · Nature Communications]

Reviewer comments, first round-

Reviewer #1 (Remarks to the Author):

The authors used complete blood counts data from two large studies to derive a novel measure of biological aging which they named dynamic morbidity index (DMI). They present evidence that DMI changes with age, correlates with frailty, and predict onset of age-related mortality as well as longevity. Age related changes in DMI are particularly important in validating it as a meaningful "biomarker" or "clock" of aging and avoiding inherent limitations of cross-sectional data. Analysis of variance of DMI and modeling its changes over time was shown to have interesting implications in terms of early life anti-aging interventions, effects of external and internal stressors, and the biological limit of lifespan. Some of the predictions agree with previous assessment of eliminating (or curving) a particular disease on life expectancy.

Several details of this very interesting and well-written paper may require the authors' attention, as listed below.

Abstract:

"quantification of aging process" is a somewhat awkward way of describing DMI. "a measure," "quantitative measure (or: marker)," or "a tool for quantitative assessment might be better.

Describing database as "sufficiently large" is somewhat unclear. Should it be "allowing meaningful statistical assessment (or a desired statistical power)," or simply "large?"

Introduction:

Since the derivation of DMI and interpretation of results refer to Gompertz mortality law, it might be of interest to mention some limitations of Gompertz's analysis and suggested modifications (for example, publications of J.J. Koopman and R. Westendorp).

For the benefit of readers who are not statistically/mathematically savvy, it would help to add brief definitions of some terms. For example, auto-correlation time, diverging of recovery, random walk, the attraction basin.

Finally, I could not find information identifying the diagnostic lab which provided some of the data.

Andrzej Bartke

Reviewer #2 (Remarks to the Author):

The authors show that with age, the temporal autocorrelation and variance of blood-count indicators increase. They argue that this is an indicator of decreasing resilience, and extrapolate the change to predict a theoretical maximum age for humans.

I greatly enjoyed reading this work. I am not a medical specialist, but from my interest in studying the resilience of complex systems the approach taken by the authors just makes a lot of sense. I think this an innovative and important contribution to the way we quantify health.

I happily recommend publication, and have just a few suggestions for the authors:

1) As far as I can see the authors only show a trend of their indicators (variance and autocorrelation) with age. It would of course be nice to see if the indicators are also predictive for the risk of all-cause-death. The data may currently not allow that, but in the discussion it could be useful to indicate follow-up research that could elaborate on the possibilities.

2) The authors seem unaware of a recent body of literature suggesting variance and autocorrelation as generic indicators on resilience related to human health as well as other

complex systems. It could be useful to link their work to some of this literature as it may strengthen the points they make and show broader implications. Below are a few examples (not that you have to cite all of this, but just as a pointer to facilitate exploring this connection).

Best regards,

Marten Scheffer

1. Gijzel SM, et al. (2018) Dynamical indicators of resilience in postural balance time series are related to successful aging in high-functioning older adults. *The journals of gerontology. Series A, Biological sciences and medical sciences*.
2. Gijzel SMW, et al. (2017) Dynamical Resilience Indicators in Time Series of Self-Rated Health Correspond to Frailty Levels in Older Adults. *Journals of Gerontology - Series A Biological Sciences and Medical Sciences* 72(7):991-996.
3. Olde Rikkert MGM, et al. (2016) Slowing Down of Recovery as Generic Risk Marker for Acute Severity Transitions in Chronic Diseases. *Critical Care Medicine* 44(3):601-606.
4. Scheffer M, et al. (2018) Quantifying resilience of humans and other animals. *Proceedings of the National Academy of Sciences*:201810630.
5. Scheffer M, et al. (2009) Early-warning signals for critical transitions. *Nature* 461(7260):53-59.
6. Scheffer M, et al. (2012) Anticipating critical transitions. *Science* 338(6105):344-348.
7. Dakos V, Carpenter SR, van Nes EH, & Scheffer M (2015) Resilience indicators: Prospects and limitations for early warnings of regime shifts. *Philosophical Transactions of the Royal Society B: Biological Sciences* 370(1659):20130263.

Reviewer #3 (Remarks to the Author):

This paper takes an interesting approach to the study of resilience over the lifespan: to evaluate the dynamics of CBC measurements over time. The authors' identification of the need to better understand "the relationship between the linear physiological state dynamics and the exponentially increasing morbidity, frailty, and mortality observed during aging" is a particularly salient question.

There are, however, a number of concerns with the current paper:

- 1) Its review of the aging literature is superficial. There is an enormous body of research leveraging epidemiological cohorts on aging, considerably much of which has evaluated longitudinal evolution of biomarkers over time. Little to none has been acknowledged. Similarly, there has been a great deal of recent activity in the area of "Geroscience," driven by sentinel papers on the "hallmarks" and "pillars" of aging. An important recent paper seeking to develop a biomarker of aging (Belsky and colleagues), and an active recent literature on physical resilience in older adults, that the authors have not cited. The literature on frailty has only been highly selectively cited. It is essential to anchor the current work more solidly in the context of this gerontologically rich body of work.
- 2) The authors label a comorbidity count as "frailty." This is unacceptable: There is fairly wide consensus in the gerontological literature that "frailty" and multimorbidity are not synonymous. The count should be identified as a "comorbidity" or "multimorbidity" count.
- 3) Very serious concern: There is no motivation and insufficient evidence that CBC count can be considered as a sensitive or specific biomarker of aging, rather than a marker of chronic disease or multimorbidity. Stronger evidence to motivate this (possibly already extant) or justify this is needed.
- 4) Very serious concern: The data analytic findings are insufficiently justified because they are very highly model dependent, and no evidence has been provided to substantiate that the model "fits" the data adequately. At the very least, evidence of fit of the proportional hazards analysis

and of the modeling used to derive the recovery rate is needed. It is not clear that even this will suffice, however, because the conclusions derived from analyses depicted in Figure 3 rely on large extrapolations. In Figure 3A, such an extrapolation relies on only a few data points, and in Figure 3B, the fit of the hypothesized curves to the data is quite poor. It is unclear that this concern can be addressed with the authors' data.

5) The data do not clearly substantiate that the co-clustering analysis identified "two dynamic subsystems." Intercorrelations within the immune subsystem appears quite modest. A PCA (with evidence as to the number of dimensions of shared variance) might better address the authors' aim than the raw correlation matrix.

6) The choice to regress mortality on the CBC elements independently of age at CBC measurement is troubling. Associations reflective of age and/or chronic disease are then incorporated into the DMI in ways that can't easily be adjusted in subsequent analyses. One would wish to know whether CBC elements predict mortality independently of age in an analysis in which the age contribution is fully (and flexibly—not by merely a linear term) adjusted first, and that these account for appreciable net concordance in the concordance index: If not, their value as an aging biomarker is called into question. If they do not predict mortality independently in an analysis also adjusting for comorbidity count, it becomes questionable whether these markers reflect more than chronic disease type or severity. (Once this evidence is provided, there is no objection to then considering the age dynamics of the DMI as the authors have proposed.) The particularly high associations with emphysema and heart failure are suspicious in this regard: They merit discussion.

7) Serious concern: The authors discarded samples with missing data for any of the CBC components. This could result in a highly selective sample—so selective that the ability of the remaining data to inform aging dynamics becomes suspect. The amount of data discarded must be reported. Multiple imputation arguably would be a preferable way to handle marker-level missingness.

8) Minor concern: Some details of the analysis are unclear. Did the authors employ a single (baseline?) CBC measurement in the mortality analyses, or were these time varying? Was the competing mortality risk for the disease onset analyses addressed in any way? Were the c-statistics cross validated (understanding that there was a very large sample size—but see concern #9)?

9) Supplement Table 2 reveals a feature of concern: Neither Haemoglobin nor Red blood cell count independently predicts mortality. The reviewer suspects that these two variables are highly collinear and either should be summarized and entered as a single variable, or that one or the other should be included but not both. (There is a similar, if lesser concern, regarding all red blood cell-related variables except distribution width.) In this case: The algorithm for producing the DMI could result in a highly variable score, which in turn might cloud all the findings pertaining to DMI variability.

Karen Bandeen-Roche

Answers to reviewers' comments:

Reviewer #1 (Remarks to the Author):

The authors used complete blood counts data from two large studies to derive a novel measure of biological aging which they named dynamic morbidity index (DMI). They present evidence that DMI changes with age, correlates with frailty, and predict onset of age-related mortality as well as longevity. Age related changes in DMI are particularly important in validating it as a meaningful “biomarker” or “clock” of aging and avoiding inherent limitations of cross-sectional data. Analysis of variance of DMI and modeling its changes over time was shown to have interesting implications in terms of early life anti-aging interventions, effects of external and internal stressors, and the biological limit of lifespan. Some of the predictions agree with previous assessment of eliminating (or curving) a particular disease on life expectancy.

Several details of this very interesting and well-written paper may require the authors' attention, as listed below.

Abstract:

“quantification of aging process” is a somewhat awkward way of describing DMI. “a measure,” “quantitative measure (or: marker),” or “a tool for quantitative assessment might be better.

Based on all the reviewers' comments we agree with the comment and decided to change the terminology. DMI is a function of the physiological indices and hence is rather the organism state variable. Thus, we suggest the new name – the dynamic organism state indicator (DOSI). The “definition” disconnects the quantity from immediate medical or biological associations and better complies with the terminology used in related earlier works.

Describing database as “sufficiently large” is somewhat unclear. Should it be “allowing meaningful statistical assessment (or a desired statistical power),” or simply “large?”

We dropped or clarified such “fuzzy” definitions as good as we could throughout the updated version. We meant, of course, that the dataset is to be large enough to allow meaningful statistical assessment. A finer point in our study is that the dataset must be longitudinal (that is to provide multiple measurements from the same person) at time intervals not too far apart to allow for robust inference of the resilience. We added an overview of previous works in this direction based on data from, e.g., Framingham Heart Study, Dunedin Multidisciplinary Health, and Development Study. In most cases, however, the observations are too far apart - often separated by several years and hence the time between the measurements greatly exceeds the organism state autocorrelation time reported here. This is why, to the best of our understanding, the relation of the organism state recovery rate and mortality has remained largely elusive.

Introduction:

Since the derivation of DMI and interpretation of results refer to Gompertz mortality law, it might be of interest to mention some limitations of Gompertz's analysis and suggested modifications (for example, publications of J.J. Koopman and R. Westendorp).

There are known limitations to Gompertzian fits to survival data. We got involved in this discussion a while ago (*Tarkhov, Menshikov, Fedichev. "Strehler-Mildvan correlation is a degenerate manifold of Gompertz fit." Journal of theoretical biology 416 (2017): 180-189; <https://www.sciencedirect.com/science/article/pii/S0022519317300176>*). The takeaways are a) it's a bad idea to fit log-transformed quantities; b) it is hard to establish at the same time an effect on the rate of aging and the age-independent proportional hazard factor.

In the present work, we do not aim at building the best mortality model from the data. Our goal is to identify a collective variable, built from multiple physiological indices and associated with the risk of death in the strongest possible way (given the available data) and then investigate its dynamic properties.

To solve the problem, we used the Cox proportional hazards model, taking only the physiological state variables (such as CBC) as inputs (no explicit age!). As the result, we do not rely on or imply Gompertz or any other specific form of the base hazard function dependence on chronological age. The approach was proposed by David Cox in the early 1970-s and since then has been used extensively (note 20k+ citations of the paper) in problems ranging from clinical trials analysis to predicting failure of mechanical systems. Only a small number of those problems concerned aging and involved biological systems governed by Gompertz mortality law.

For the benefit of readers who are not statistically/mathematically savvy, it would help to add brief definitions of some terms. For example, auto-correlation time, diverging of recovery, random walk, the attraction basin.

We have substantially rewritten the abstract and the main text of the manuscript to clarify our approach and the model assumptions. We brought forward the math and explanations from the supplementary to improve the presentation flow of the manuscript. We hope that it helps.

Finally, I could not find information identifying the diagnostic lab which provided some of the data.

We used the anonymized set of longitudinal CBC measurements obtained from patients visiting InVitro clinic centers in Russia. Also, we added physical activity data collected using Fitbit fitness wristbands by users of our app. We inserted the necessary datasets' descriptions in the main text. We are planning to make the data available for everyone once the manuscript is accepted.

Andrzej Bartke

Reviewer #2 (Remarks to the Author):

The authors show that with age, the temporal autocorrelation and variance of blood-count indicators increase. They argue that this is an indicator of decreasing resilience, and extrapolate the change to predict a theoretical maximum age for humans.

I greatly enjoyed reading this work. I am not a medical specialist, but from my interest in studying the resilience of complex systems the approach taken by the authors just makes a lot of sense. I think this an innovative and important contribution to the way we quantify health.

I happily recommend publication, and have just a few suggestions for the authors:

1) As far as I can see the authors only show a trend of their indicators (variance and autocorrelation) with age. It would of course be nice to see if the indicators are also predictive for the risk of all-cause-death. The data may currently not allow that, but in the discussion, it could be useful to indicate follow-up research that could elaborate on the possibilities.

Thank you for noting. The data is indeed scarce and it is hard to make a solid statement. Based on the physical picture we speculate that the variance and the autocorrelation time are indeed another two independent biomarkers of aging and should be independent risk factors. In the updated version we included these arguments in the Discussion section.

2) The authors seem unaware of a recent body of literature suggesting variance and autocorrelation as generic indicators on resilience related to human health as well as other complex systems. It could be useful to link their work to some of this literature as it may strengthen the points they make and show broader implications. Below are a few examples (not that you have to cite all of this, but just as a pointer to facilitate exploring this connection).

Thank you for pointing this out, we added more discussion and references to provide a more comprehensive review of resilience.

Best regards,

Marten Scheffer

- 1. Gijzel SM, et al. (2018) Dynamical indicators of resilience in postural balance time series are related to successful aging in high-functioning older adults. The journals of gerontology. Series A, Biological sciences and medical sciences.**
- 2. Gijzel SMW, et al. (2017) Dynamical Resilience Indicators in Time Series of Self-Rated Health Correspond to Frailty Levels in Older Adults. Journals of Gerontology - Series A Biological Sciences and Medical Sciences 72(7):991-996.**

3. Olde Rikkert MGM, et al. (2016) Slowing Down of Recovery as Generic Risk Marker for Acute Severity Transitions in Chronic Diseases. *Critical Care Medicine* 44(3):601-606.
4. Scheffer M, et al. (2018) Quantifying resilience of humans and other animals. *Proceedings of the National Academy of Sciences*:201810630.
5. Scheffer M, et al. (2009) Early-warning signals for critical transitions. *Nature* 461(7260):53-59.
6. Scheffer M, et al. (2012) Anticipating critical transitions. *Science* 338(6105):344-348.
7. Dakos V, Carpenter SR, van Nes EH, & Scheffer M (2015) Resilience indicators: Prospects and limitations for early warnings of regime shifts. *Philosophical Transactions of the Royal Society B: Biological Sciences* 370(1659):20130263.

Reviewer #3 (Remarks to the Author):

This paper takes an interesting approach to the study of resilience over the lifespan: to evaluate the dynamics of CBC measurements over time. The authors' identification of the need to better understand "the relationship between the linear physiological state dynamics and the exponentially increasing morbidity, frailty, and mortality observed during aging" is a particularly salient question.

Thank you for your very helpful comments and questions, we made our best effort to address your concerns in the revised version of the manuscript. We hope that we were able to make multiple improvements to our manuscript.

There are, however, a number of concerns with the current paper:

1) Its review of the aging literature is superficial. There is an enormous body of research leveraging epidemiological cohorts on aging, considerably much of which has evaluated longitudinal evolution of biomarkers over time. Little to none has been acknowledged. Similarly, there has been a great deal of recent activity in the area of "Geroscience," driven by sentinel papers on the "hallmarks" and "pillars" of aging. An important recent paper seeking to develop a biomarker of aging (Belsky and colleagues), and an active recent literature on physical resilience in older adults, that the authors have not cited. The literature on frailty has only been highly selectively cited. It is essential to anchor the current work more solidly in the context of this gerontologically rich body of work.

We expanded our literature overview by adding references to reviews on the “hallmarks” of aging and studies of longitudinal data. We also expanded the discussion of recent literature on physical resilience in older adults.

We also added a review of papers on the study of aging in longitudinal data in the Discussion section. The bottom line is that most of the current longitudinal studies provide insufficient sampling rate, that is most of the time the subsequent samples are years apart. As it turns out (and revealed by our analysis), this is inappropriate to study resilience. According to our findings the time between the measurements typically greatly exceeds the organism state autocorrelation time.

2) The authors label a comorbidity count as “frailty.” This is unacceptable: There is fairly wide consensus in the gerontological literature that “frailty” and multimorbidity are not synonymous. The count should be identified as a “comorbidity” or “multimorbidity” count.

This is also corrected. In the new version, we use the compound morbidity index (CMI) instead of the frailty index where appropriate. The log-mortality risk predictor is now the dynamics organism state indicator (DOSI) to reflect its role as an organism state variable, rather than an accurate measure of frailty. Extreme values of DOSI are still associated with extreme frailty index and multiple morbidities, but we decided to avoid semantics and possible confusion by disentangling DOSI and frailty in its names.

3) Very serious concern: There is no motivation and insufficient evidence that CBC count can be considered as a sensitive or specific biomarker of aging, rather than a marker of chronic disease or multimorbidity. Stronger evidence to motivate this (possibly already extant) or justify this is needed.

Over the last few years, researchers identified multiple quantitative biomarkers of the aging process, most notably using DNA methylation (aka methylation-age, *Horvath, Steve. "DNA methylation age of human tissues and cell types." *Genome Biology* 14.10 (2013): 3156; Hannum, Gregory, et al. "Genome-wide methylation profiles reveal quantitative views of human aging rates." *Molecular cell* 49.2 (2013): 359-367*), transcriptomics (*Peters, Marjolein J., et al. "The transcriptional landscape of age in human peripheral blood." *Nature communications* 6.1 (2015): 1-14*), the gut microbiome (*Odamaki, Toshitaka, et al. "Age-related changes in gut microbiota composition from newborn to centenarian: a cross-sectional study." *BMC microbiology* 16.1 (2016): 1-12*), or even 3D face shapes (*Xia, Xian, et al. "Three-dimensional facial-image analysis to predict heterogeneity of the human aging rate and the impact of lifestyle." *Nature Metabolism* (2020): 1-12*). The discovery of the best aging clock (or clocks) is still underway. The consensus emerges that biological clocks can be trained using different biological signals (from molecular levels to digitalized physical activity streams) and those trained to predict mortality perform better than models of chronological age.

The most relevant to our analysis model, the PhenoAge (Levine, Morgan E., et al. "An epigenetic biomarker of aging for lifespan and healthspan." *Aging (Albany NY)* 10.4 (2018): 573), is a log-mortality risk predictor based on clinical biochemistry and CBC. PhenoAge is used in clinical trials and was used as a target to train the risk-based DNA-methylation clock — Grim age (Lu, Ake T., et al. "DNA methylation GrimAge strongly predicts lifespan and healthspan." *Aging (Albany NY)* 11.2 (2019): 303).

In this work, we use two types of biological signals. The choices are mostly the result of necessity. Our goal was to get our hands on the data which is big enough (many samples) and is sampled at a frequency that is high enough for analysis of autocorrelation properties.

The purely CBC based log-risk predictor proposed in our study and referred to as DOSI turned out to be not so bad. Of course, the higher levels of risk (DOSI) were associated with multimorbidity. On the other hand, higher levels of risk (DOSI) in people with no diagnosed diseases were associated with increased risks of future chronic diseases and mortality in UKB and NHANES. We also checked that in people with no diseases, the elevated levels of DOSI were reversibly associated with unhealthy lifestyles, such as smoking.

In our recent work, (Ying, Kejun, et al. "Genetic and Phenotypic Evidence for the Causal Relationship Between Aging and COVID-19." *medRxiv* (2020) <https://www.medrxiv.org/content/10.1101/2020.08.06.20169854v1>), we used PhenoAge and DOSI (along with the log-transformed average physical activity) to show that elevated DOSI and reduced physical activity levels are significantly associated with elevated risks of a non-chronic disease (COVID) in patients with and free of chronic diseases separately. The CBC-based DOSI performed fairly well on par with PhenoAge (see Fig. 2 therein)

To summarize this point: the choice of CBC is that of a necessity. The log-risk model from CBC can be improved by adding biochemistry and a lot more variables, but we did not have a chance to get that kind of data with a sufficient sampling rate. The CBC-based model is almost as good as the most popular PhenoAge when it comes to predicting future health risks in non-frail (disease-free) populations and hence may be a proxy of aging (rather than multimorbidity).

We are citing all the relevant papers (other than our new preprint) in the updated version of the manuscript.

4) Very serious concern: The data analytic findings are insufficiently justified because they are very highly model dependent, and no evidence has been provided to substantiate that the model "fits" the data adequately. At the very least, evidence of fit of the proportional hazards analysis and the modeling used to derive the recovery rate is needed. It is not clear that even this will suffice, however, because the conclusions derived from analyses depicted in Figure 3 rely on large extrapolations. In Figure 3A, such an extrapolation relies on only a few data points, and in Figure 3B, the fit of the

hypothesized curves to the data is quite poor. It is unclear that this concern can be addressed with the authors' data.

The evidence of the fit of the proportional hazards analysis is provided in the Materials and Methods section (Supplementary information). The Cox proportional hazards model was trained using NHANES 2015 Public-Use Linked Mortality data. CBC data and mortality linked follow-up available for 4059 2NHANES participants aged 18–85 y.o. was used. We trained the Cox model using the data of participants aged 40–85 y.o. (11731 male and 12076 female) with 3792 recorded death events during follow-up until the year 2015 (1999–2014 surveys). CBC components and the biological sex label were used as covariates.

The model was well-predictive of all-cause mortality and yielded a concordance index value of $CI = 0.68$ in NHANES. The model was subsequently validated in an independent dataset from UK Biobank (UKB) and produced a similar value of $CI = 0.66$ in UKB (samples collected 2007–2011, 216250 male and 255223 female participants aged 39–75 y.o., 13162 recorded death events during follow-up until the year 2016).

Next, let us address the issue of “highly model dependent findings”.

To prove the universality of our results, in the revised version we complemented our analysis with the auto-correlation properties of another organism state indicator, closely associated with the mortality and morbidity, that is (the negative logarithm of) averaged physical activity. Fortunately, we acquired a large longitudinal dataset from wearable devices and hence were able to demonstrate the key features of the universality of the proposed model. We observed that:

- The longest auto-correlation time in the physical activity time series is the same as the auto-correlation time in CBC measurements time series (both measured by the autocorrelation decay time in Fig 3C).
- The auto-correlation time increases (and the auto-correlation rate decreases) in both time series of the physical activity and CBC measurements at the same rate in cohorts of individuals of increasing age
- The recovery rate decreases and the dynamic range of the organism state fluctuations increases in such a way that the resilience vanishes and the fluctuations diverge at the same age. The outcome does not depend on whether you measure blood markers or physical activity levels.

Such universality of large scale fluctuations in the vicinity of breaking point was expected by us and supported by the analysis of two independent signals.

The general argument works as follows: due to the criticality, in the proximity of the bifurcation point the organism state dynamics is dominated by a single degree of freedom associated with the system decay into the unstable phase. This scenario is closely related to the Landau-Ginzburg phase transition theory and is commonly invoked in the modeling of transient behaviors in complex systems (as highlighted, e.g., in M. Scheffer's works). Close to such

phase transition points the dynamics acquire universal features that are independent of the dynamic properties of subsystems.

One of the most important consequences of criticality is the dramatic dimensionality reduction, that is the long-term dynamic properties of a complex system can be effectively described by very few (most often the single) dynamic variables (see a recent discussion in *Barzel, Baruch, and Albert-László Barabási. "Universality in network dynamics." Nature physics 9.10 (2013): 673-681* <https://www.ncbi.nlm.nih.gov/pmc/articles/PMC3852675/>). Such a composite feature is a macroscopic property of a non-equilibrium system (the order parameter). In the proximity of the phase transition, the dynamic range of the order parameter fluctuations is large, and, at the same time, the dynamics of the order parameter is slow (compared to the “microscopic” time scales). The corresponding phenomena are known as critical fluctuations and critical slowing down and are hallmarks of transient states, see any of M. Scheffer’s references.

The universality and the dimensionality reduction means that most of the fluctuations observed in any of the sub-systems of the aging organism are associated with the dynamics of the same feature (the order parameter) and hence should be characterized by the same fluctuation and auto-correlation properties. This theoretical concept is also known as “enslavement principle” (as in [https://en.wikipedia.org/wiki/Synergetics_\(Haken\)](https://en.wikipedia.org/wiki/Synergetics_(Haken))) and is a cornerstone of synergetics and the main paradigm in self-organization studies.

Finally, we do not claim (and no available data lets us state) that there is the actual divergence of the organism state fluctuations at the specific age. We only observe that the organism state dynamics is consistent with that in the proximity of the critical point and that it leads to increasing cross-correlations between the physiological indices, auto-correlation times and risks of death and diseases. The physical attainability of the critical point is an open and yet not so practically an important issue. We can only be sure (and that is what we observe in life tables) is that very few individuals have a chance of even getting close. This is because in the real world we find quite strong fluctuations that drive the transition into the unstable (terminal) state well below 100+ years old.

The new results are now part of the revised manuscript and, in our opinion, provide sufficient evidence in favor of the universality of the organism state dynamics late in life. The association of the order parameter with mortality (through the proportional hazards model) helps to relate the critical fluctuations and the loss of resilience to the disintegration of the organism. The proposed physical picture may help understand how slow (linear) remodeling of the resilience with age may be the driving force behind the fast (exponential) growth of risks of death and diseases.

The novel observation in our work is that not only the levels of specific markers are indicative of human health. Such an approach is dominant in the vast corpus of works devoted to biological age determination. As suggested in some earlier works, the auto-correlation properties of the organism state indices are important features of the organism’s state (and health) on their own.

This is not a totally trivial finding since according to the same analysis, mice have no physiological resilience on the whole organism dynamics level almost from the beginning of their life and disintegrate exponentially (see Identification of a blood test-based biomarker of aging through deep learning of aging trajectories in large phenotypic datasets of mice).

We hope that the revised version of the manuscript reflects these (apparently correct) views in the right way.

5) The data do not clearly substantiate that the co-clustering analysis identified “two dynamic subsystems.” Intercorrelations within the immune subsystem appears quite modest. A PCA (with evidence as to the number of dimensions of shared variance) might better address the authors’ aim than the raw correlation matrix.

We removed the raw correlation matrix figure. Instead, we provided the PCA of the whole dataset and presented the “aging trajectories” capturing a coordinated change of the physiological indices in the course of the organism development and aging in updated Figure 1A. On top of that, we added a plot of age-dependence of DOSI in the whole age range and established its concordance with a theoretical model for ontogenetic growth in novel Figure 1B. We believe that the newly presented results together provide stronger evidence for DOSI being an (or strongly associated with) the key organism-level dynamic state variable associated with the development and aging, rather than merely an output of a machine learning model.

We hope that this part of the manuscript is now improved.

6) The choice to regress mortality on the CBC elements independently of age at CBC measurement is troubling. Associations reflective of age and/or chronic disease are then incorporated into the DMI in ways that can’t easily be adjusted in subsequent analyses. One would wish to know whether CBC elements predict mortality independently of age in an analysis in which the age contribution is fully (and flexibly—not by merely a linear term) adjusted first, and that these account for appreciable net concordance in the concordance index: If not, their value as an aging biomarker is called into question. If they do not predict mortality independently in an analysis also adjusting for comorbidity count, it becomes questionable whether these markers reflect more than chronic disease type or severity. (Once this evidence is provided, there is no objection to then considering the age dynamics of the DMI as the authors have proposed.) The particularly high associations with emphysema and heart failure are suspicious in this regard: They merit discussion.

CBC elements do predict health risks and are associated with unhealthy lifestyles independent of the number of chronic diseases and age. We demonstrate this by performing Cox-proportional hazards models significance tests using the “biological age acceleration” (the difference between the CBC-derived DOSI level and its age- and sex- matched cohort average; this is exactly the adjustment for chronological age and sex beyond the linear term). This “biological age acceleration” is in fact age- and gender-adjusted DOSI in the cohort-wise

manner and not by merely a linear regression. The results can be found in Table S2 and Fig 2C of the revised version.

The issue of including or not including the explicit age (or its powers) into the proportional hazards model though appearing to be only a technical issue raises an important “conceptual” point.

Indeed, if the goal of your work is to establish the best predictor of mortality, then a Cox proportional hazards model with explicit age is a good model and can arguably produce superior results. This is the way how, e.g., a popular PhenoAge model is constructed by S. Hovarth and M. Levine.

In the present work we, however, aimed to study the dynamic laws governing the evolution of the organism state in relation to aging. In such case adding the explicit age (that is time) in the consideration would amount to assuming that the organism state dynamics depends on time explicitly and on the system state variables, rather than on the system state variables only.

The “explicit time” hypothesis would require an additional dependence on time since birth and in such a way of the existence of a special clock “providing” the time signal to every subsystem of the organism. Such a clock could be, in principle, discovered and possibly disrupted in an experiment that would lead to a very dramatic life-extension effect.

We suggest that there is no such a clock and the role of the clock variable is played by the order parameter itself. The biological time is thus an emergent property, that is the biological age is dynamically acquired from the fluctuations of the order parameter. This is more “natural” (in the specific sense) since it does not require building a specific mechanism for timekeeping.

Time will tell which vision is right and we left this discussion mostly out of the manuscript.

We used both types of log-hazards model in our studies and none was better than the other. Using explicit time (age) has a major disadvantage: age is the single most important risk factor of death and diseases and hence the log-hazard predictor of a model with explicit age is almost a linear function of age with some small variable part at any age. This may effectively hinder the proper analysis of any nonlinear age-dependence in the data.

The organism state dynamics is richer than that. As a teaser, in the revised version of the manuscript, we included the organism state dynamics trajectory in the whole age range. Now Figure 1 shows the clear life-staging with the first stage describing the development. As pointed out in G. West’s work, the ontogenetic growth is governed by the resource (metabolic output) competition between the tissue growth and repair (published in Nature and Science, see <https://vsavage.faculty.biomath.ucla.edu/Assets/PDF/OntogeenticGrowth.pdf>).

You could never effectively describe this phase of life using a model with explicit age. Now, in the revised version of the manuscript, we have references to West’s works and have both the early and the late-life dynamic features described in the same framework invoking only universal (that is model free) language.

As for the particularly strong association of DOSI with emphysema, it is known to be specifically strongly associated with increased neutrophil counts. We added this to the main text with appropriate citation.

7) Serious concern: The authors discarded samples with missing data for any of the CBC components. This could result in a highly selective sample—so selective that the ability of the remaining data to inform aging dynamics becomes suspect. The amount of data discarded must be reported. Multiple imputation arguably would be a preferable way to handle marker-level missingness.

The fraction of participants with missing CBC data was less than 0.0035 (i.e. 0.35%) in each of the three studied datasets. Taking into account the large number of samples in the study we considered it appropriate to discard those samples. We added the explanations data to the Materials and Methods section.

8) Minor concern: Some details of the analysis are unclear. Did the authors employ a single (baseline?) CBC measurement in the mortality analyses, or were these time varying? Was the competing mortality risk for the disease onset analyses addressed in any way? Were the c-statistics cross validated (understanding that there was a very large sample size—but see concern #9)?

We used a proportional hazards model as a method to obtain an organism-level variable associated with aging. As such we left the time-varying CBC values intact intentionally to obtain the age-dependent dynamics of the organism state variable (DOSI) associated with aging and mortality.

We did not perform any more sophisticated mortality analysis. Table S2 shows that DOSI is associated with mortality and morbidity independently of chronological age and morbidity status.

9) Supplement Table 2 reveals a feature of concern: Neither Haemoglobin nor Red blood cell count independently predicts mortality. The reviewer suspects that these two variables are highly collinear and either should be summarized and entered as a single variable, or that one or the other should be included but not both. (There is a similar, if lesser concern, regarding all red blood cell-related variables except distribution width.) In this case: The algorithm for producing the DMI could result in a highly variable score, which in turn might cloud all the findings pertaining to DMI variability.

Some of the individual features comprising the vector of CBC measurements are highly collinear and hence the log-linear risk model regression coefficients belonging to the individual variables cannot be interpreted individually (small variation in the input data may produce large deviations in the coefficients without changing the output result). Table 2 was a poor way to report the

model properties and reflected our poor judgement. We thank the reviewer for pointing this out and dropped the Table and removed this confusing data from the text.

The CBC measurements are very noisy and given the very large number of samples vs. the number of CBC measurements, the risk model appears to be very stable even without explicit regularization. We trained the model in NHANES and cross-validated the model in an independent dataset (UKB), see our response to Question (4).

Finally, we produced the variance analysis for a single variable, the negative logarithm of physical activity (steps/day). The single variable is known to be associated with all cause mortality and its variance increased (and the inverse variance diverged and thus hinting at the same critical point) in the same way as the variance of DOSI. We consider it as evidence of the general nature of reported results and as such should not be dramatically affected by the choice of a specific set of studied biological or physiological parameters

Reviewer comments, second round-

Reviewer #1 (Remarks to the Author):

I feel that the authors responded to the comments I had and that the paper is improved by the revisions made in response to comments/criticism from all reviewers. I believe the described approach to analyzing resilience is original and valuable.

Andrzej Bartke

Reviewer #3 (Remarks to the Author):

I appreciate the effort the authors have invested in reviewing their paper. In many respects the manuscript is improved from the previous version.

There remains two serious concerns, two moderate concerns, and a set of minor concerns:

1) Serious - The DOSI was cross-validated in UKB but appears not to have undergone any cross-validation (CV) in NHANES (no internal CV/ jackknife). Yet, many of the paper's findings are from NHANES. The paper's findings would be strengthened by reproducing the findings using an internally cross-validated version of the NHANES score (or, by splitting NHANES into randomly chosen training and test sets--spanning the whole age range--and reproducing the findings with the NHANES test set). Otherwise, there is endogeneity with respect to individuals' death times. This is particularly problematic for incidence analyses where future information will have been incorporated to predict these events.

2) Very substantial extrapolation remains in Figure 3.

3) Moderate: Values for persons younger than 40 were based on the model for persons older than 40. Thus, it is possible that the PC path shown in Figure 1A reflects lack of fit rather than biology. A model incorporating all ages (potentially with nonlinearity to accommodate potentially different relationships at younger versus older ages) would have been preferable if relationships for persons younger than 40 are to be studied.

4) Moderate: The authors have declined to present an argument in the paper for biologic plausibility of CBC as a measure of biologic aging. This seems easily addressable--a few sentences would suffice.

5) Minor: There are a few apparent mis-statements (typo variety) and ambiguous terminology uses in the paper. For example, Fig 2A y-axis is labeled as "Excess DMI" when the legend and narrative refer to it as "Excess DOSI". There is reference to "age-matching" when in fact it appears that these findings were from models with age adjustment.

6) Minor: The index has been relabeled as "CMI", but the authors continue to refer to its high ranges as "most frail." If the authors feel strongly about such a labeling, a sentence of justification is warranted.

Point-by-point response to reviewers' comments:

Reviewer #1 (Remarks to the Author):

I feel that the authors responded to the comments I had and that the paper is improved by the revisions made in response to comments/criticism from all reviewers. I believe the described approach to analyzing resilience is original and valuable.

We would like to thank the reviewer for comments that helped us to improve the text of the manuscript.

Reviewer #3 (Remarks to the Author):

I appreciate the effort the authors have invested in reviewing their paper. In many respects the manuscript is improved from the previous version.

We would like to thank the reviewer for very detailed comments that helped us to improve the presentation and interpretation of results in the manuscript. Please find our responses to your questions below. Also, we made multiple minor corrections unrelated to your questions with the goal to improve the presentation (also marked in red in the main text).

There remains two serious concerns, two moderate concerns, and a set of minor concerns:

1) Serious - The DOSI was cross-validated in UKB but appears not to have undergone any cross-validation (CV) in NHANES (no internal CV/ jackknife). Yet, many of the paper's findings are from NHANES. The paper's findings would be strengthened by reproducing the findings using an internally cross-validated version of the NHANES score (or, by splitting NHANES into randomly chosen training and test sets--spanning the whole age range--and reproducing the findings with the NHANES test set). Otherwise, there is endogeneity with respect to individuals' death times. This is particularly problematic for incidence analyses where future information will have been incorporated to predict these events.

We agree with the reviewer. We followed the rightful suggestion and recalculated and replotted all the results in the manuscript. To perform cross-validation in NHANES, we have now split the NHANES dataset randomly into training and test sets with the ratio 50/50. We used only the training set samples to train the DOSI model. The performance of the model is assessed using the Concordance Index (CI). The calculations produced CI=0.68 and CI=0.67 in NHANES training and test sets, respectively, and CI=0.65 in UKB. We added the cross-validation results in the Materials and Methods section (lines 1050-1070, changes are highlighted in red).

The cross-sample correlation of the newly produced DOSI with the previous version was $r = 0.98$. All results and figures are now updated according to the new DOSI, although there were no significant changes in numbers and plots.

2) Very substantial extrapolation remains in Figure 3.

We re-read the manuscript and double checked that in all 3 occurrences in the Results section, the association of the critical point with the specific age or age range involves references to extrapolation.

We also clarified our position on the character of the extrapolation in an extra paragraph (also involving “extrapolation” language) in the discussion (lines 688-705, highlighted in red):

“Our study provides evidence suggesting that late in life the organism state dynamics is dominated by features that originate from the proximity of the critical point, corresponding to the vanishing resilience. The exact parameters, such as maximum lifespan, are the results of extrapolations yielding the estimate in the range of 100-150 years. The questions of whether the critical point corresponds to a specific age or even achievable along a realistic trajectory, is not too practical: due to the presence of strong stochastic forces, most individuals escape the attraction basin, lose the resilience and disintegrate into states corresponding to chronic diseases well before reaching the ultimate age. Hence the extrapolation may serve to establish the upper bound on attainable age or the limiting lifespan.”

We also added a reference to a related work (lines 714-719, highlighted in red):

“We also note, that very late in life, as the probability of the loss of resilience increases, so should the deviations from Gompertz mortality law. A recent careful analysis of human demographic data supports this argument and yields an estimate for limiting lifespan of 138 years (Podolskiy, Dmitriy I., et al. "The landscape of longevity across phylogeny." *bioRxiv* 2020).”

Note also the improved wording in the following paragraph “The semi-quantitative description of human aging and morbidity proposed here “should work well long before the maximum age” (lines 721-722, highlighted in red) and belongs to a class of phenomenological models.”

Hopefully these better stated arguments clarify our position well.

3) Moderate: Values for persons younger than 40 were based on the model for persons older than 40. Thus, it is possible that the PC path shown in Figure 1A reflects lack of fit rather than biology. A model incorporating all ages (potentially with nonlinearity to accommodate potentially different relationships at younger versus older ages) would have been preferable if relationships for persons younger than 40 are to be studied.

The basis for Fig 1A is Principal Components Analysis (PCA), which is an unsupervised technique and is used here mainly for data visualization. When we perform PCA, we do not rely on any particular model, and the lack of fit may only occur due to the lack of data. NHANES provides a very substantial number of individuals younger than 30 years, and hence we believe that the features we observe in Figure 1A are real. Interestingly, qualitatively the same PCA

“trajectory” and life-staging features can be observed in apparently unrelated PCA of features extracted from human physical activity patterns in Figure 2 of (Pyrkov et al., *Aging (Albany NY)* 10.10 (2018): 2973). We refer to this result in the Discussion section (lines 461-466, highlighted in red)

Our view is that Figure 1A explains why it is reasonable to train a linear Cox proportional hazards model using the data from individuals older than approximately 40 y.o. We agree that a full non-linear model would perform better, although it would be much harder to do, since the mortality in younger individuals is negligible and no good model can be built using the available data.

Using DOSI for individuals younger than 40 years old is, of course, an extrapolation, although not a bad one. Indeed, the age dynamics of DOSI demonstrated a good fit to the theoretical prediction for a quantity associated with the organism development in G. West theory of growth. The dynamics of DOSI is, according to G. West, highly non-linear. The dependence of the physiological indices on DOSI appears to be linear and hence the theoretical curve fits the NHANES data fairly well.

We provided the fit and a brief explanation of the West's theory as an extra argument suggesting the biological significance of the features on the PCA plot. Such fit is highly unlikely to be merely a coincidence by chance.

4) Moderate: The authors have declined to present an argument in the paper for biologic plausibility of CBC as a measure of biologic aging. This seems easily addressable--a few sentences would suffice.

The reason to use CBC in our study was a) the wide availability of CBC measurements in datasets and b) the known association of CBC parameters with both age and mortality (survival).

For example, neutrophil-to-lymphocyte ratio (NLR) increases with age (Zhou et al., 2019, *Aging Clinical and Experimental Research*, 1-9; Liu et al., 2015, *Orl*, 77(2), 109-116) and is predictive of survival in many studies (Lu et al., 2017, *BioMed research international*, vol. 2017; Zhang et al., 2020, *Aging (Albany NY)*, 12(3), 2428; Orditura et al., 2016, *ESMO open*, 1(2); Ozyurek et al., 2017, *Asian Pacific Journal of Cancer Prevention: APJCP*, 18(5), p.1417; Cataudella et al., 2017, *Journal of the American Geriatrics Society*, 65(8), pp.1796-1801).

Another example is red blood cell distribution width (RDW) which has also been reported to increase with age (Lippi et al., 2014, *Clinical Chemistry and Laboratory Medicine (CCLM)*, 52(9), pp.e197-e199; Hoffmann et al., 2015, *Clinical Chemistry and Laboratory Medicine (CCLM)*, 53(12)) and to be predictive of survival (Arbel et al., 2014, *Thrombosis research*, 134(5), 976-979; Seyhan et al., 2013, *Journal of Chronic Obstructive Pulmonary Disease*, 10(4), 416-424)

We added relevant references in Introduction (lines 68-72, highlighted in red).

5) Minor: There are a few apparent mis-statements (typo variety) and ambiguous terminology uses in the paper. For example, Fig 2A y-axis is labeled as "Excess DMI" when the legend and narrative refer to it as "Excess DOSI". There is reference to "age-matching" when in fact it appears that these findings were from models with age adjustment.

Corrected. Thank you for pointing at this typo.

6) Minor: The index has been relabeled as "CMI", but the authors continue to refer to its high ranges as "most frail." If the authors feel strongly about such a labeling, a sentence of justification is warranted.

As suggested by the reviewer, we added the justification to use CMI as a proxy to Frailty Index in text (see lines 196-208, highlighted in red):

CMI can be viewed as a convenient proxy to the Frailty Index introduced in (Blodgett et al., Arch Gerontol Geriatr 60.3 (2015): 464-470), that is a composite marker, depending on the prevalence of 46 health deficits. Unlike the Frailty Index, CMI requires only the variables that are available simultaneously in NHANES and UKB. In NHANES, among individuals aged 40 and older, the correlation between the Frailty Index and CMI was pretty high (Pearson's $r=0.64$). Therefore, we accept semi-quantitative correspondence between CMI and Frailty Index and categorize UKB and NHANES participants in cohorts of individuals of increasing level of frailty according to CMI.

Reviewer comments, third round-

Reviewer #3 (Remarks to the Author):

Thank you for your dedicated work on this manuscript. The reviewer admires your passion and perseverance and has no further comments to be addressed.

Karen Bandeen-Roche